# Evidence of Autonomic Dysfunction in Patients with Relapsing-Remitting Multiple Sclerosis: Heart Rate Variability and Cardiovascular Parameters

**Liudmila Gerasimova-Meigal** [1,*], **Ilya Sirenev** [2] and **Alexander Meigal** [1]

1   Department of Human and Animal Physiology, Petrozavodsk State University, 185910 Petrozavodsk, Russia; meigal@petrsu.ru
2   Department of Physiology and Pathophysiology, Republican Hospital V. A. Baranov, 185019 Petrozavodsk, Russia; i.sirenev@yandex.ru
*   Correspondence: gerasimova@petrsu.ru; Tel.: +7-911-402-9907

**Abstract:** This study was aimed at evaluation of autonomic dysfunction in patients with multiple sclerosis (MS) by means of time- and frequency-domain parameters of heart rate variability (HRV) and conventional cardiovascular tests (deep breathing (DB) and active orthostatic test (AOT)). The study group enrolled 32 patients with the relapsing-remitting MS (17 m, 15 f, aged $29 \pm 4.9$ years, disease duration $4.2 \pm 2.7$ years, EDSS scores less than 3.0 and 26 subjects in good health (HC, 15 m, 11 f, aged $30.1 \pm 2.7$ years). In the MS group, at rest the variability of heart rate was decreased in comparison to the HC group seen by time- (SDNN, RMSSD, pNN50, CV, $p < 0.01$) and frequency-domain (TP, HF, LF, $p < 0.05$) parameters, what was indicative of the general decrease of the autonomic neurogenic control of the heart rate, both sympathetic and parasympathetic. The functional tests (DB and AOT) showed reduced cardiovascular reactivity in the MS group. Additionally, the cardio-respiratory synchronization was impaired in the MS group at rest and DB. The severity of HRV deficit in the MS group correlated with the activity of MS. In conclusion, the comprehensive assessment of time- and frequency-domain HRV parameters studied with functional tests provides better insight to understanding autonomic dysfunction in subjects with relapsing-remitting MS.

**Keywords:** multiple sclerosis; heart rate variability; autonomic dysfunction; sympathetic nervous system; parasympathetic nervous system





## 1. Introduction

Autonomic disorders are typical of multiple sclerosis (MS). Some subclinical impairments of the autonomic nervous system, along with clinically isolated syndrome, manifest early in MS and are evident in more than 80% MS patients [1–5]. Inflammatory-degenerative processes in the hypothalamus, anterior commissure, internal capsule of brain, brainstem autonomic centers and nervous pathways in the spinal cord are considered as a morphological basis of autonomic dysfunction in MS [5–7]. Autonomic disorders in MS are represented by a wide spectrum of syndromes reported in many studies [4,6,8]. According to them, the cardiovascular dysfunction is apparent in 42–63% of patients with MS, sexual dysfunction–in 34–85%, impaired pupillary reactions in 26–60%, gastrointestinal disorders in 40–81%, abnormal sudomotor reactions in 26–94%, thermoregulatory disorders in 60–80%, and pelvic disorders in 92%.

The cardiovascular dysfunction in MS appears as fast fatigue during physical activity, orthostatic hypotension, dizziness, postural orthostatic tachycardia, nausea, syncope, and left ventricular failure, which can lead to pulmonary edema [2,4,6,8,9]. In more than half patients with MS, subclinical cardiovascular dysfunction in the form of altered heart rate variability (HRV) is diagnosed even prior to the onset of clinically evident neurological syndromes [3]. Moreover, impaired function of the cardiovascular system is often detected

regardless of the initial neurological deficit in MS [10]. As the autonomic regulation of the cardiovascular system reflects the autonomic control of the whole organism, its changes are considered as predictive signs of MS progression [4,11–13].

Despite extensive studies of autonomic dysfunction in MS, including evaluation of HRV, there is still no consensus, which part of the autonomic nervous system–sympathetic and/or parasympathetic is the most impaired at different stages of MS [2]. Some studies evidence that in MS the sympathetic autonomic control is deficient, while others report on the deficit of the parasympathetic control [7,13]. Videira G. [7] noted combined impairment of the sympathetic and parasympathetic autonomic regulation, which was detected in 45.5% of patients with MS. In our earlier study, in MS patients we have reported on enhanced sensitivity to cold and reduced sympathetic control of sudomotor function evaluated with skin sympathetic responses (SSR) [14–16]. Therefore, we hypothesized that the role of the sympathetic and parasympathetic system can be better elucidated with a combination of the HRV parameters and some additional cardiovascular tests.

In sum, this study was aimed at evaluation of the autonomic dysfunction in patients with relapsing-remitting MS with help of HRV time- and frequency-domain parameters, and conventional cardiovascular tests.

## 2. Experimental Section

### 2.1. Participants

The study enrolled 54 people (32 men, 22 women) aged 22 to 40 years. After comprehensive verbal explanation, all participants signed an informed consent form to participate. Study protocol was approved by joint Ethic committee of the Ministry of Health care of the Republic of Karelia and Petrozavodsk State University (30, 16.06.2014).

The study group (MS group) consisted of 28 patients (17 men and 11 women, aged above 18 years) with a definitive diagnosis of relapsing-remitting MS based on McDonald criteria [5,17]. The average age of the study group was $29.6 \pm 4.2$ years, and the average duration of the disease was $4.2 \pm 2.7$ years. Eighteen patients with MS had an exacerbation of MS verified by MRI, and 14 patients had a remission of MS. The neurological status of the patients in the study group was rather uniform in respect with the following criteria: all patients had similar localization of foci of demyelination according to MRI of the brain, involvement of no more than three functional systems according to Kurtzke [18,19], the mean Expanded Disability Status Scale (EDSS) score less than 3.0 points, the arm function score from 0 to 1 point on the Arm Index scale, and the neurological status on the SCRIPPS scale from 85 to 100 scores. According to MRI, juxta-/intracortical and periventricular foci of demyelination were found in all MS patients, infratentorial–in 18 MS patients, and the spinal ones (on the cervical or thoracic level)–in 6 MS patients. The rate of various neurological manifestations in MS patients is shown in Table 1.

**Table 1.** Frequency of involvement by Functional System in the studied MS group according to J. Kurtzke [19].

| System | Number of MS Patients |
|---|---|
| Pyramidal | 63% |
| Cerebellar | 59% |
| Brain stem | 44% |
| Sensory | 22% |
| Bowel and Bladder | 19% |
| Visual | 22% |
| Cerebral | 9% |
| Other | 6% |

Since MS patients had constantly taken disease-modifying treatments, the study enrolled only patients who did not require drugs with a notable effect on the autonomic regulation and/or cardiac function.

The control group included 26 apparently healthy individuals (HC, 15 men, 11 women), comparable in age with the study group, without chronic somatic and neurological diseases. Their average age was $30.1 \pm 2.7$ years. The anthropometric characteristics of participants of both groups are presented in Table 2.

**Table 2.** Anthropometric characteristics of the participants at the time of their inclusion in the study.

| Parameter | MS:M (*n* = 17) | MS:W (*n* = 11) | HC:M (*n* = 15) | HC:W (*n* = 11) |
|---|---|---|---|---|
| Body Mass, kg | $75.7 \pm 9.9$ | $59.6 \pm 17.4$ | $77.8 \pm 9.4$ | $57.6 \pm 4.4$ |
| Height, m | $1.78 \pm 0.04$ | $1.68 \pm 0.10$ | $1.78 \pm 0.07$ | $1.60 \pm 0.06$ |
| BMI | $24.0 \pm 3.1$ | $21.2 \pm 6.2$ | $24.5 \pm 3.3$ | $22.9 \pm 2.9$ |
| Metabolic rate, kcal | $1816 \pm 177$ | $1361 \pm 177$ | $1817 \pm 2$ | $1328 \pm 74.8$ |

Abbreviations: MS, Multiple Sclerosis group; HC, healthy control group; M, men; W, women; BMI, body mass index.

### 2.2. Outcome Measures

The HRV measurements were performed under laboratory conditions (room temperature 22–24 °C, $22.9 \pm 1.1$ °C in average, humidity 50–60%, air velocity less than 0.1 m/s). All tests were performed in the morning after the subject stood for 30 min indoors to stabilize skin temperature.

The electrocardiogram (ECG) was recorded in the standard lead II, and further HRV analysis was performed with "VNS-Spectr" device (Neurosoft Ltd., Ivanovo, Russia). First, the ECG was recorded at rest in supine position at spontaneous respiratory rate of 12–17 per min (in average $14 \pm 4$ per min) for 5 min. Then, the subjects were instructed to perform deep breathing (DB) at a respiratory rate of 6 per minute. During DB, ECG was recorded over 15 breathing cycles. Finally, the active orthostatic test (AOT) was performed, at which the ECG was recorded for 5 min in standing position.

The analysis of HRV was performed in accordance with international Standards of measurement, physiological interpretation, and clinical use [20,21]. All ECG records were visually inspected for stationarity, and all artifacts were corrected. The HRV was evaluated with the time- and frequency-domain parameters. Time-domain HRV parameters included the heart rate (HR), R-Rmin, R-Rmax, standard deviation (SDNN), root mean squared difference (RMSSD) and proportion of successive intervals greater than 50 ms (pNN50%) of normal RRi (NN). Frequency-domain HRV parameters included the total power (TP) spectrum of RRi [20], power spectrum at very low (VLF; <0.04 Hz), low (LF; 0.04–0.15 Hz), and high frequency bands (HF; 0.15–0.40 Hz), the LF/HF ratio, and spectrum structure (% VLF,% LF,% HF, LF n.u, HF n.u.). At AOT, the coefficient K30:15 (the quotient of RRi around 30th beat by that at the 15th) after standing was estimated. Cardiorespiratory interaction was assessed by the respiratory arrhythmia coefficient (Cra), as the ratio of R-Rmax to R-Rmin [21,22].

Systolic and diastolic blood pressure (SBP and DBP, respectively) and heart rate (HR) were measured with UA-705 digital tonometer (A&D Company LTd, Japan) at R and AOT.

### 2.3. Statistical Analysis

Data were analyzed using the IBM SPSS Statistics 21.0 software (IBM, Armonk, NY, USA). Differences between MS and HC groups were assessed with non-parametric Mann-Whitney and Spearman tests. Time- and frequency-domain HRV parameters were tested for their dependence on the inflammatory activity of MS by ranking according to activity of MS (0 for the HC group, 1 for MS patients in remission and 2 for MS patients in exacerbation) with Spearman test. The results were considered significant at $p$ less than 0.05.

## 3. Results

### 3.1. HRV Measurements at Rest

The results of time- and frequency-domain HRV measurements in MS and HC groups are presented as Me (25%; 75%) in Table 3. In the MS group, at rest time-domain HRV pa-

rameters (SDNN, RMSSD, pNN50, CV) were lower in comparison with the HC group, what indicates on reduced parasympathetic control of the heart rhythm. Additionally, low value of Cra in the MS group was indicative of the decreased cardiorespiratory synchronization in MS patients compared to HC (see Table 3).

**Table 3.** Time- and frequency-domain HRV parameters.

| Test | Parameter | HC | MS | Significance [1] |
|---|---|---|---|---|
| Rest | HR, per minute | 64 (61; 68) | 70 (62; 76) | $p < 0.05$ |
| | R-R min, ms | 785 (700; 818) | 750 (669; 815) | n.s. |
| | R-R max, ms | 1103 (1046; 1173) | 1043 (906; 1105) | $p < 0.05$ |
| | RRNN, ms | 939 (880; 987) | 860 (794; 968) | $p < 0.05$ |
| | SDNN, ms | 51 (42; 67) | 39 (30; 49) | $p < 0.01$ |
| | RMSSD, ms | 43 (35; 60) | 29 (21; 47) | $p < 0.01$ |
| | pNN50, % | 22.7 (10.3; 35.4) | 5.6 (1.0; 27.9) | $p < 0.01$ |
| | CV, % | 5.55 (4.38; 7.38) | 4.44 (3.67; 5.36) | $p < 0.01$ |
| | TP, $ms^2$ | 2529 (1766; 4598) | 1551 (1047; 2527) | $p < 0.01$ |
| | VLF, $ms^2$ | 832 (437; 1188) | 544 (377; 870) | n.s. |
| | LF, $ms^2$ | 755 (419; 1713) | 577 (310; 702) | $p < 0.05$ |
| | HF, $ms^2$ | 894 (446; 1587) | 407 (232; 998) | $p < 0.05$ |
| | LF/HF | 1.00 (0.43; 1.93) | 1.07 (0.55; 1.70) | n.s. |
| | % VLF | 32.9 (19.5; 41.3) | 37.7 (27.6; 52.1) | n.s. |
| | % LF | 32.2 (17.3; 43.5) | 27.9 (23.1; 36.8) | n.s. |
| | % HF | 33.6 (18.7; 45.6) | 25.9 (18.5; 41.3) | n.s. |
| | LF, n.u. | 49.95 (29.90; 65.60) | 51.65 (35.33; 62.88) | n.s. |
| | HF, n.u. | 50.05 (34.40; 70.10) | 48.35 (37.13; 64.68) | n.s. |
| | Cra | 1.38 (1.32; 1.63) | 1.32 (1.24; 1.38) | $p < 0.05$ |
| DB | HR, per minute | 69 (64; 76) | 71 (65; 77) | n.s. |
| | R-R min, ms | 670 (618; 738) | 688 (638; 768) | n.s. |
| | R-R max, ms | 1133 (1021; 1278) | 1055 (901; 1196) | $p < 0.05$ |
| | RRNN, ms | 853 (781; 934) | 852 (781; 918) | n.s. |
| | SDNN, ms | 107 (81; 133) | 78 (51; 114) | $p < 0.01$ |
| | RMSSD, ms | 72 (57; 100) | 48 (27; 73) | $p < 0.01$ |
| | pNN50, % | 34.6 (23.9; 46.1) | 25.8 (6.5; 40.4) | $p < 0.05$ |
| | CV, % | 12.70 (9.53; 16.33) | 9.34 (6.72; 12.33) | $p < 0.01$ |
| | Cra | 1.65 (1.47; 1.87) | 1.45 (1.32; 1.66) | $p < 0.01$ |
| AOT | HR, per minute | 83 (79; 86) | 94 (84; 99) | $p < 0.05$ |
| | R-R min, ms | 613 (579; 626) | 550 (518; 595) | $p < 0.01$ |
| | R-R max, ms | 855 (804; 955) | 803 (716; 865) | $p < 0.05$ |
| | RRNN, ms | 719 (685; 771) | 635 (608; 716) | $p < 0.01$ |
| | SDNN, ms | 47 (37; 63) | 42 (29; 51) | n.s. |
| | RMSSD, ms | 21 (17; 33) | 18 (12; 23) | $p < 0.05$ |
| | pNN50, % | 2.8 (0.8; 12.8) | 1.8 (0.1; 3.4) | $p < 0.05$ |
| | CV, % | 6.45 (5.44; 8.03) | 5.99 (4.53; 7.74) | n.s. |
| | TP, $ms^2$ | 3188 (1907; 5013) | 2309 (1217; 3903) | n.s. |
| | VLF, $ms^2$ | 1105 (717; 2046) | 989 (749; 1655) | n.s. |
| | LF, $ms^2$ | 1131 (836; 2414) | 931 (409; 2074) | n.s. |
| | HF, $ms^2$ | 313 (145; 656) | 178 (93; 289) | $p < 0.05$ |
| | LF/HF | 4.26 (2.69; 7.66) | 6.60 (3.81; 9.34) | n.s. |
| | % VLF | 36.1 (27.1; 49.4) | 50.9 (38.4; 63.0) | $p < 0.01$ |
| | % LF | 47.8 (36.8; 64.4) | 41.3 (26.4; 55.1) | n.s. |
| | % HF | 9.7 (7.4; 15.3) | 6.3 (4.2; 9.9) | $p < 0.01$ |
| | LF, n.u. | 80.95 (72.88; 88.43) | 86.80 (79.20; 90.35) | n.s. |
| | HF, n.u. | 19.05 (11.58; 27.13) | 13.20 (9.68; 20.80) | n.s. |
| | Cra | 1.42 (1.32; 1.53) | 1.43 (1.26; 1.52) | n.s. |
| | K30:15 | 1.35 (1.23; 1.53) | 1.24 (1.19; 1.32) | $p < 0.01$ |

Abbreviations: MS, Multiple Sclerosis group; HC, healthy control group; R, rest condition; DB, deep breathing test; AOT, active orthostatic test. [1] The significance is based on Mann-Whitney test and Spearman correlation.

The frequency-domain HRV parameters (TP and its LF and HF components) were significantly decreased (see Table 3), what reflects general deficit of the neurogenic regulation of HR. The ratio of main frequency domains in the MS group was 38%–28%–26% (VLF > LF > HF), which corresponds to predominance of central ergotropic (sympathetic) and humoral metabolic factors in the regulation of HR. Most frequency-domain HRV parameters in the MS group significantly differed from those obtained in the HC group. Furthermore, in the HC group, the frequency domain components were almost equal 33%–32%–34% (VLF–LF–HF), which means a well-balanced autonomic regulation of the HR.

### 3.2. HRV Measurements at DB

At DB, in the MS group the increase of the general variability of HR was not so notable as that in the HC group. Lower values of time-domain HRV parameters (SDNN, RMSSD, pNN50, CV) and Cra corresponded with a deficit of the parasympathetic control on the heart rate regulation (see Table 3).

### 3.3. HRV Measurements at AOT

At AOT, blood pressure has only insignificantly increased in both groups, and no orthostatic hypotension was found. In the MS group an increase of HR at AOT was more evident than in the HC group (Table 4). In general, at AOT the variability of HR was lower than in healthy individuals (see Table 3). In MS, the spectrum structure of HRV was characterized by notable increase of the VLF–domain what presumably corresponded with humoral metabolic factors, and by marked decrease of the HF–domain, what corresponded with deficit of the parasympathetic control of the heart function. The spectrum structure in MS patients was as follows: 51%–41%–6% (VLF >> LF >> HF). In the HC group the spectrum structure was 48%–36%–10% (LF > VLF > HF), which corresponds to adequate, balanced neurogenic control of the HR. The MS group was characterized by low K30:15 values, which is supposed to be a result of decreased baroreceptor reflex activity [23,24]. In the MS group it was $1.27 \pm 0.15$, and in the HC group–$1.42 \pm 0.23$ in average ($p < 0.01$, according to the Mann–Whitney and Spearman criteria).

**Table 4.** Hemodynamic parameters in the studied groups.

| Test | Parameter | HC | MS |
|------|-----------|-----|-----|
| Rest | SBP, mm Hg | $114 \pm 9$ | $112 \pm 9$ |
|      | DBP, mm Hg | $67 \pm 7$ | $68 \pm 5$ |
|      | HR, per minute | $64 \pm 9$ | $69 \pm 9$ * |
| AOT  | SBP, mm Hg | $122 \pm 13$ | $118 \pm 12$ |
|      | DBP, mm Hg | $83 \pm 10$ | $80 \pm 8$ |
|      | HR, per minute | $83 \pm 9$ | $92 \pm 13$ ** |

Abbreviations: MS, Multiple Sclerosis group; HC, healthy control group; R, rest condition; AOT, active orthostatic test. *, $p < 0.05$, **, $p < 0.01$, the significance of differences from HC is based on Mann-Whitney test.

### 3.4. HRV Measurements in MS of Different Activity

Patients in exacerbation of MS had a more markedly decreased variability of HR seen from the time- and frequency-domain parameters in comparison with MS patients in remission of the disease and HC (Figures 1 and 2). Such tendency was the characteristic either for the rest state or the functional tests (DB and AOT).

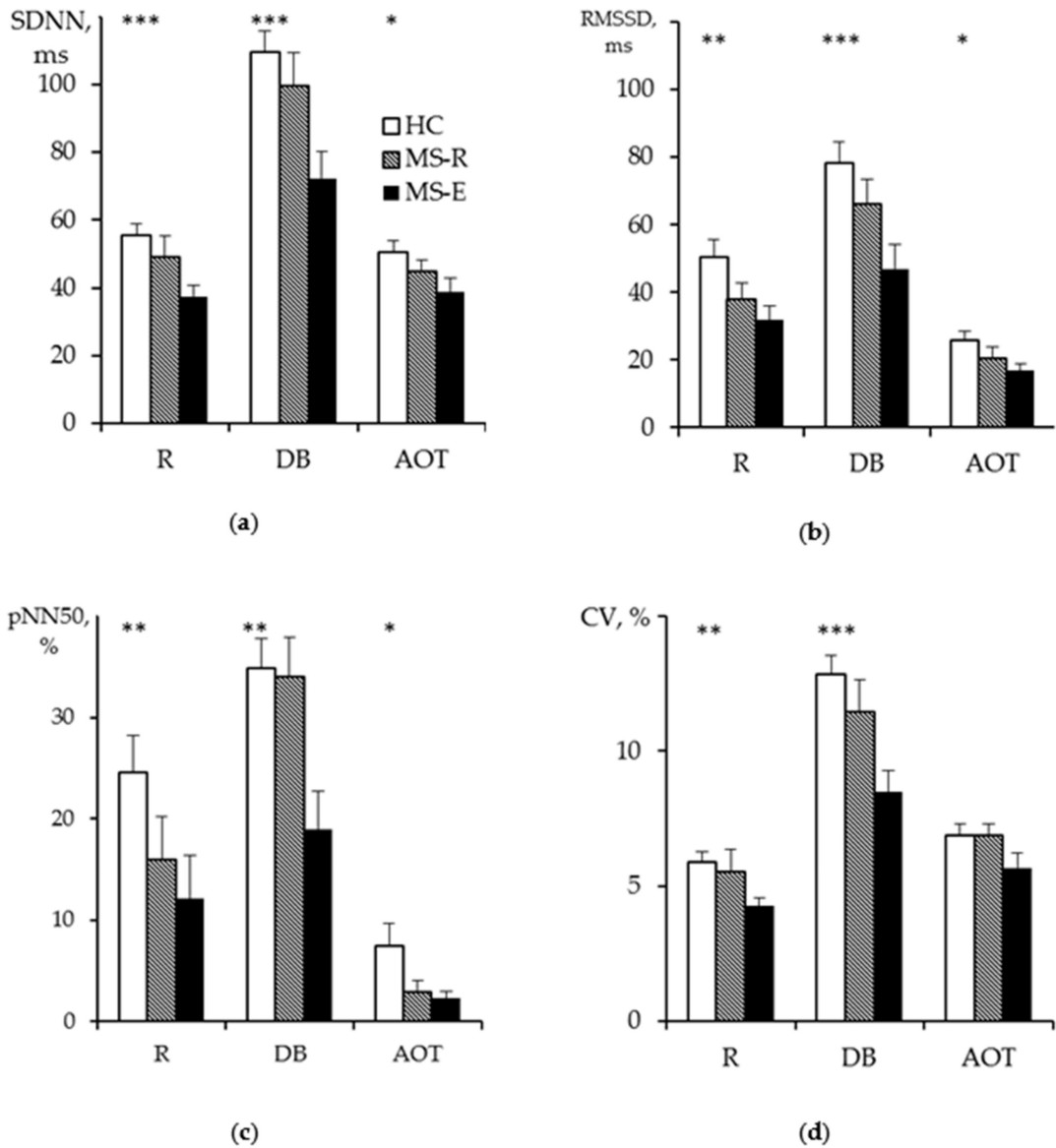

**Figure 1.** Time-domain HRV parameters in MS groups with different disease activity: (**a**) SDNN; (**b**) RMSSD; (**c**) pNN50; (**d**) CV. Abbreviations: HC, healthy control group; MS-R, Multiple Sclerosis remission group; MS-E, Multiple Sclerosis exacerbation group; R, rest condition; DB, deep breathing test AOT, active orthostatic test. *, $p < 0.05$, **, $p < 0.01$, ***, $p < 0.001$, the significance of Spearman correlation.

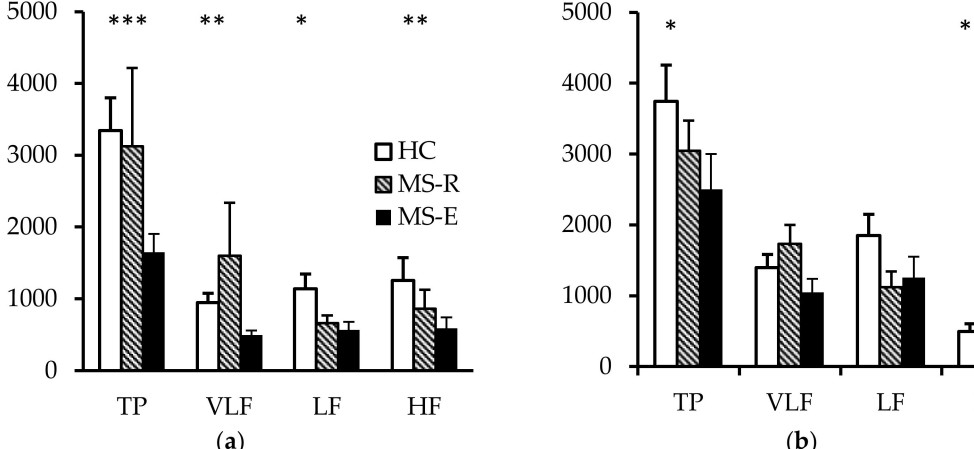

**Figure 2.** Frequency-domain HRV parameters in MS groups with different disease activity: (**a**) at rest condition; (**b**) at active orthostatic test. X–axis, frequency bands; Y-axis, power, ms$^2$. Abbreviations: HC, healthy control group; MS-R, Multiple Sclerosis remission group; MS-E, Multiple Sclerosis exacerbation group. * $p < 0.05$, ** $p < 0.01$, *** $p < 0.001$, the significance of Spearman correlation.

## 4. Discussion

The original idea of this study held that application of HRV parameters, both of time- and frequency domain, in combination with the cardiovascular tests would allow better detecting of impairments in the autonomic nervous control in MS. The methodological approach and major findings of the study are presented on Scheme 1. In our study, we applied most of the available time- and frequency-domain parameters of HRV, and such approach showed that in patients with relapsing-remitting MS these parameters allowed documenting reduction of both the sympathetic and parasympathetic neurogenic control of the heart rhythm. These results are in a good line with the data, which demonstrated combined deficit of the sympathetic and parasympathetic autonomic nervous system [7,15].

The impairment of the parasympathetic mechanisms of the autonomic control in MS in our study was clearly seen as decreased values of time-domain HRV parameters. Correspondingly, when only a limited set of HRV characteristics, mostly the time-domain, is used, one can conclude only about the parasympathetic control of HR [13,25]. As for the sympathetic mechanism of the autonomic control, our study showed that it could be reliably assessed with the frequency-domain HRV analysis. The function of the sympathetic nervous system is usually corresponded to the LF-domain [20,21]. Many authors report on the decreased LF-band in MS patients and relate this to the reduced sympathetic control [3,11,26]. Activity of the sympathetic nervous system could be directly evaluated by the sudomotor function with help of SSR assessment [7,15,27]. Our earlier results showed the reduced SSR in MS patients [16], which is in a good line with the reduced LF-band found in this research.

Autonomic tests, such as DB, AOT, and some others, are widely used to evaluate the cardiovascular reactivity in MS patients. Some studies reported on the autonomic dysfunction at the earlier stage of MS measured with autonomic tests [2,3,7]. In contrast, Crnošija L. et al. [1] did not find impaired cardiovascular reactivity measured with cardio-respiratory synchronization in relapsing-remitting MS.

Our study showed the reduced cardiovascular reactivity in MS patients when HRV parameters were evaluated during DB and AOT. We found it useful to detect autonomic dysfunction in relapsing-remitting MS. At DB, reduced parasympathetic reactivity was detected by decreased HRV seen as lowered time-domain parameters (SDNN, RMSSD, pNN50, CV), and less cardio-respiratory synchronization (Cra). At AOT, in MS patients, baroreceptor HRV modulation was reduced seen as decreased K30:15 value. However, MS patients did not present orthostatic intolerance at AOT. This corresponds with the studies [2,7], and is most probably due to generally low EDSS and shorter duration of

the disease. This might explain the absence of overt sympathetic autonomic deficit and orthostatic hypotension in MS patients as it was already reported [4,6,9,10]. At AOT, MS patients had greater increase of HR and predominance of VLF-band in HRV spectrum what was a marker of humoral/metabolic derived effects on HR. Surprisingly, LF-domain at AOT in MS patients did not significantly change as it was observed by Habek M. et al. [3].

Some authors found the association of various autonomic dysfunctions with the stage, duration, or activity of the disease [6,10,11], while others did not reveal such dependency [7]. The loss of sympathetic reactivity is the characteristic of the earlier stage of MS and it correlates with inflammation and MS clinical activity [1–3,5]. Respectively, the deficit of parasympathetic regulation is closely related to the progression of disability in patients with MS [14].

In our study, the most evident autonomic dysfunction was observed in patients in exacerbation of the relapsing-remitting MS. With help of time- and frequency-domain HRV measurements, we found both sympathetic and parasympathetic autonomic deficit. It is generally accepted that impairment of the autonomic nervous system in MS develops as a neuroinflammatory cycle of pathogenesis due to altered immune response [2]. Sympathetic and parasympathetic dysfunction differently manifests over the course of MS [6]. The inflammatory activity in MS correlates mostly with sympathetic autonomic deficit [2,3,6,14]. Therefore, the parasympathetic deficit could not be so evident in exacerbation of MS. Nonetheless, in our study, a more notable parasympathetic deficit was found in MS patients in exacerbation, which was attributed as a consequence of MS [14]. In contrast, the study [25] based on time-domain HRV measurements (SDNN and RMSSD) reported no difference between MS patients and HC and between predefined subgroups with various MS course.

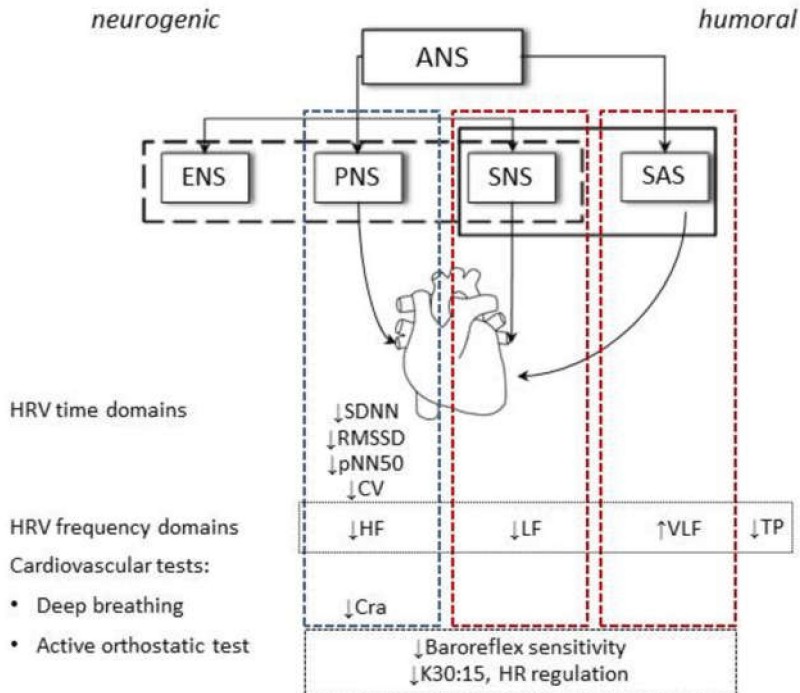

**Scheme 1.** The methodological approach to evaluation of autonomic dysfunction in MS. Abbreviations: ANS, the autonomic nervous system; ENS, the enteric nervous system; PNS, the parasympathetic nervous system; SNS, the sympathetic nervous system; SAS, the sympathetic adrenomedullary system.

## 5. Conclusions

The major outcome of the study was that the assessment of autonomic deficit in patients with relapsing-remitting MS should include both time- and frequency HRV domains

measurements performed both at rest and at cardiovascular tests. This would have allowed the fullest characterizing of both sympathetic and parasympathetic autonomic dysfunction in MS. Our study also showed that the combined use of time- and frequency domain HRV analysis allows differentiating autonomic dysfunction between remission and exacerbation in the relapsing-remitting MS.

**Author Contributions:** Conceptualization, L.G.-M., I.S. and A.M.; methodology, L.G.-M. and A.M.; validation, L.G.-M. and I.S.; formal analysis, L.G.-M. and I.S.; investigation, L.G.-M. and I.S.; resources, A.M.; data curation, I.S.; writing—original draft preparation, L.G.-M. and I.S.; writing—review and editing, L.G.-M. and A.M.; visualization, L.G.-M.; supervision, L.G.-M.; project administration, A.M.; funding acquisition, A.M. All authors have read and agreed to the published version of the manuscript.

**Funding:** This research was funded by the Ministry of Science and Higher Education of the Russian Federation, grant number 0752-2020-0007.

**Institutional Review Board Statement:** The study was conducted according to the guidelines of the Declaration of Helsinki and approved by joint Ethic committee of the Ministry of Health care of the Republic of Karelia and Petrozavodsk State University (№30, 16.06.2014).

**Informed Consent Statement:** Informed consent was obtained from all subjects involved in the study.

**Data Availability Statement:** The data presented in this study are available on request from the corresponding author.

**Conflicts of Interest:** The authors declare no conflict of interest.

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
