# Peer review of "Evidence of Autonomic Dysfunction in Patients with Relapsing-Remitting Multiple Sclerosis: Heart Rate Variability and Cardiovascular Parameters"

_pathophysiology, doi:10.3390/pathophysiology28010002_

Round 1
Reviewer 1 Report
This study reports autonomic evaluation in patients with MS versus healthy controls
These are results from frequency-domain and time-domain
1) There methodology and calculations are confusing
For example, frequency analysis should be presented at rest and orthostatic but not during DB.
The only parameter worthwhile register in DB is the I:E ratio
2) DB, orthostatism 30:15 ratio is part of Ewing battery. 30:15 is not reported. Other components of EB were left out? why? Any orthostatic hypotension
3) Videira et al documented similar findings and the importance of frequency analysis to detect autonomic dysfunction in MS. How are these findings novel? I suggest to restructure the presentation of results and focus on rest and orthostatic and the findings of exacerbation
4) should document MS with brain stem ou spinal involvement MS without it (more probable to have autonomic dysfunction)
5) Conclusions: recommend SSR but this was not evaluated. Keep on the results.
Overall, the discussion can be shortened
Reviewer 2 Report
It is a simple but straightforward paper. Sample size is limited but the authors make the point the raise in their introduction.
I have some comments listed hereafter
- The authors should consider in their intro also data coming from a recently published paper about almost 600 patients treated with Fingolimod. They will find analogies specifically as far as spectral HRV analyses concerns.
- Time domain analyses are usually computed from long term ECG recording. Their value in short term ECG recordings is less validated. The opposite is true for spectral HRV.This should be discussed
- Line 128: Why not to use normalized units instead of % as recommended in international guidelines?
- Line 129: what is “the functional reserve of the organism”? When HR is high at baseline there might be a ceiling effect but certainly not here
- Table 3. LF/HF is identical as probably it would be if LF and HF power were computed in normalized unit. This does not really diminish the meaning of the study but points to two main findings: in MS there is indeed a global loss in HRV and AOT documents an increase reflex sympathetic reactivity as documented by the HR behavior (more than HRV)
- Ln 146: VLF cannot be accounted for inshort time acquisition in which too few cycles are available to quantify power.
- Table 4. Data suggest elevated sympathetic responses to the orthostatic stress. It is worth noting, though, that the peripheral pressor response is maintained thus suggesting adequate cardiac and vascular receptorial response to NE.
